# An Anomaly Detection Algorithm Based on Ensemble Learning for 5G Environment

**DOI:** 10.3390/s22197436

**Published:** 2022-09-30

**Authors:** Lifeng Lei, Liang Kou, Xianghao Zhan, Jilin Zhang, Yongjian Ren

**Affiliations:** Computer & Software School, Hangzhou Dianzi University, Hangzhou 310018, China

**Keywords:** self-attention, ensemble learning, anomaly detection, SDN, 5G

## Abstract

With the advent of the digital information age, new data services such as virtual reality, industrial Internet, and cloud computing have proliferated in recent years. As a result, it increases operator demand for 5G bearer networks by providing features such as high transmission capacity, ultra-long transmission distance, network slicing, and intelligent management and control. Software-defined networking, as a new network architecture, intends to increase network flexibility and agility and can better satisfy the demands of 5G networks for network slicing. Nevertheless, software-defined networking still faces the challenge of network intrusion. We propose an abnormal traffic detection method based on the stacking method and self-attention mechanism, which makes up for the shortcoming of the inability to track long-term dependencies between data samples in ensemble learning. Our method utilizes a self-attention mechanism and a convolutional network to automatically learn long-term associations between traffic samples and provide them to downstream tasks in sample embedding. In addition, we design a novel stacking ensemble method, which computes the sample embedding and the predicted values of the heterogeneous base learner through the fusion module to obtain the final outlier results. This paper conducts experiments on abnormal traffic datasets in the software-defined network environment, calculates precision, recall and F1-score, and compares and analyzes them with other algorithms. The experimental results show that the method designed in this paper achieves 0.9972, 0.9996, and 0.9984 in multiple indicators of precision, recall, and F1-score, respectively, which are better than the comparison methods.

## 1. Introduction

The advancement of communication technology has altered the face of human civilization as it enters the digital information era. The advancement of information technology will have an impact on the ease of living in human civilization. With the arrival of the 5G era, human society’s level of informatization will become even higher. In comparison to 4G, the 5G network’s application scenarios will span the sectors of mobile Internet, Internet of Vehicles, and the Industrial Internet. Simultaneously, operators have set greater standards for 5G networks, including huge transmission capacity, ultra-long transmission distance, network slicing, and intelligent management and control. Among them, the software-defined network (SDN) is a new type of network design idea that intends to increase network flexibility and agility and can better fulfill the network slicing demands of 5G networks. The central concept of software-defined networking is to decouple past network architecture into the control plane and the data plane and to previous abstract network functions into applications in the network operating system in the control plane [1]. From top to bottom, the software-defined network architecture is split into the application plane, control plane, infrastructure layer, and physical device layer [2,3], as illustrated in Figure 1. The system components of the application plane include application applications and network management systems. This applies to control plane service requests made via the northbound interface provided by the SDN regulator [4]. One or more SDN controllers comprise the control plane. In the software-defined network architecture, the SDN controller serves as a bridge between the application plane and the infrastructure layer. On the one hand, the SDN controller exposes diverse programmable services to upper-layer application software via the northbound interface, and network users can flexibly formulate network policies based on actual application scenarios; on the other hand, the SDN controller constructs and maintains a global network view via the southbound interface to control and manage network devices at the infrastructure layer, and inherits the control plane functions. The infrastructure layer is made up of data-forwarding devices such as switches and routers that were abstracted into network devices. The data flow is handled in accordance with the instructions given by the SDN controller, thereby improving network device management efficiency. The physical layer includes control equipment, including field instruments, sensors, and actuators and performs duties such as information interchange between the ICS controller and field equipment. SDN has gotten much attention from people from many areas of life.

However, software-defined networks are vulnerable to cyber-attacks in the same way that traditional networks are. As previously said, SDN introduces the SDN controller, which provides unified API services for the application plane and the infrastructure layer, allowing the network to be centralized, programmable, and open. These characteristics, such as permitting mismatched network packets to be submitted to the controller to request forwarding rules, raise security issues for SDN. A network assault is frequently exhibited as anomalous traffic. The term “abnormal traffic” refers to network traffic behavior that deviates from the expected typical pattern. Server overload induced by DOS assaults, worms’ privileged access, and server attacks will result in anomalous traffic [5]. SDN network security risks primarily target the control plane, with the majority of attacks targeting the network’s controller [6]. Malicious controllers, malware, and malicious switches can all put SDN controllers at risk. The controller’s security has a direct influence on SDN security since it is the centralized decision-making entity and processing hub of SDN.

When aberrant traffic is detected, abnormal traffic detection technology monitors network traffic transmission immediately, sends an alarm, or takes active reaction steps. The real-time monitoring of SDN traffic may maintain the security, confidentiality, and integrity of SDN network information while also promoting the development and implementation of SDN technology [5]. As a result, research on intrusion detection systems in the context of SDN offers tremendous theoretical and application value for creating and upgrading SDN technology.

Hinton, Geoffrey E. et al. [7] proposed the concept of deep learning in 2006. With the continuous improvement in computer computing power and the continuous development of algorithms, deep learning algorithms that require huge computing power have attracted great attention from researchers and enterprises. Traditional detection algorithms based on traffic feature statistics and machine learning perform better when small-scale datasets and feature quantities are small. However, it still relies on the manual judgment and induction of traffic characteristics. Deep learning algorithms can calculate optimal solutions from limited data and do not require expert knowledge to find unknown and new abnormal traffic types. With large-scale datasets and many features, it can also have better performance.

We propose an abnormal traffic detection method based on the stacking method and self-attention mechanism (TSMASAM) that combines the self-attention mechanism and ensemble learning to make up for the inability of ensemble learning to learn the associations between data. First, we propose a neural network composed of a self-attention mechanism and a deep convolutional network which aims to automatically learn the correlation between traffic samples, capture the feature space’s internal structure, and provide it downstream in the form of a sample embedding task. Secondly, we design a novel stacking integration method, which aims to detect and identify abnormal network traffic by integrating the sample embedding obtained above and the inspection results of the heterogeneous base learner. Finally, we design a new loss function, which fully and comprehensively considers the basic learner’s influence on the model’s overall performance by introducing the basic learner’s loss value and the regular term composed of it and preventing the model from falling into an overfitting state.

The main contributions of this paper are as follows:We propose a neural network composed of a self-attention mechanism and a deep convolutional network, which learns from samples and converts them into sample embeddings.We propose a stacking ensemble learning method composed of the autoencoder and base learner, using the autoencoder to remove irrelevant information in the samples and the stacking method to integrate the detection results of sample embedding and the base learner.We design a novel loss function to observe the operation of the model through the introduced regularization term and base learner loss value. We use a network traffic dataset under an SDN architecture to evaluate the model’s performance. The results show that the model has a better abnormal traffic detection effect than the comparison model.

The structure of this paper is as follows: Section 2 briefly describes the research status of related work; Section 3 introduces the experimental environment, model framework, and specific design of TSMASAM; Section 4 details the experiments and performance evaluation of TSMASAM proposed in this paper. In Section 5, we conclude the paper.

## 2. Related Works

The concept of abnormal traffic detection technology began in the 1980s, which refers to a network security technology that monitors network traffic transmission, promptly issues an alarm or takes active response measures when abnormal traffic is found. Anomaly detection obtains a feature model by modeling and analyzing traffic characteristics, thereby judging whether the network traffic is normal. Anomaly detection technology can be roughly divided into: that based on traffic feature matching, based on traffic feature statistics, and based on machine learning. There are essential differences in the modeling logic of the three traffic characteristics, which bring about different detection scenarios and inspection effects. Algorithms based on traffic feature matching require professionals to analyze and summarize the characteristics of abnormal traffic and then match them with the observed traffic characteristic. The advantage of this approach is the higher accuracy of identifying known attacks. However, algorithms based on traffic feature matching rely on expert knowledge and are difficult to deal with unseen abnormal network traffic. The anomaly detection algorithm based on data statistics is based on the normal distribution of network traffic characteristics. When the observed network traffic characteristics deviation from the benchmark exceeds a certain threshold, it is regarded as abnormal network traffic. The advantage of the algorithm based on data feature statistics is that it is simple to implement and can also identify abnormal traffic in unknown networks, but it is prone to misjudgment. Algorithms based on machine learning have a stronger learning ability and can learn from incomplete traffic characteristics to abnormal traffic, however, the models generally have high computational complexity and are not suitable for high-response environments. However, with the substantial improvement in computing technology, methods based on machine learning have received attention and research from all parties.

Based on the traffic feature matching algorithm, the abnormal network traffic is known quickly, but it cannot effectively deal with the unknown abnormal network traffic. Ref. [8] proposed the model NADIR, a near real-time expert system, to replace the manual review log method. NADIR compares the network activity summarized in user profiles with expert rules that define network security policies, inappropriate or suspicious network activity, and normal network and user activity. Ref. [9] proposed an adaptive real-time intrusion detection expert system which contains a statistical subsystem to observe the normal traffic of the computer. The statistical subsystem identifies user behavior as a potential intrusion when it observes significant deviations from expected behavior. Ref. [9] maintains a knowledge base of statistical topics consisting of profiles, and updates the observed behaviors to the knowledge base daily. Before the new statistics are synchronized to the knowledge base, the previous statistics are multiplied by a decay factor to adaptively learn the behavior patterns of the observers. Ref. [10] proposed an approach to specification-based and anomaly-based intrusion detection by starting from the state machine specification of the network protocol and supplementing the state machine information with statistical information. Ref. [10] verified the effectiveness of this method on the KDD99 dataset. Furthermore, Ref. [10] uses a protocol specification to simplify the feature selection step. Ref. [11] described network intrusion detection expert system (NIDX) by combining knowledge describing the target system, the historical profiles of users’ past activities, and knowledge-based intrusion detection heuristics. NIDX classifies user activity through the UNIX system calls and then uses knowledge and heuristics about typical intrusion and attack techniques to determine whether the activity is anomalous. Ref. [12] built a method to augment domain knowledge with machine learning to create rules for intrusion detection expert systems. To this end, Ref. [12] adopted a combination of genetic algorithms and decision trees to automatically generate rules for classifying network traffic. The algorithm based on the feature matching relies on the analysis and summary of professionals and is not sufficiently flexible to operate.

The algorithm based on data statistics can quickly identify abnormal traffic and deal with unknown abnormal network traffic, but it is prone to misjudgment. A histogram-based outlier detection (HBOS) algorithm was proposed to score data in linear time. Since HBOS assumes no dependencies between features, the algorithm is technically faster than other methods but less accurate [13]. HBOS detects global outliers such as the state-of-the-art algorithm on multiple datasets but performs poorly on local outliers. Ref. [14] described the anomaly detection problem as a binary composite hypothesis testing problem and developed a model-free and model-based approach using large deviation theory. Both methods extracted a series of probability laws representing traffic patterns over different time periods and then detect anomalies by evaluating the traffic and deviations from these laws. Ref. [15] proposed a statistical signal processing technique based on mutation detection. Authors such as M. Thottan demonstrated the method’s feasibility in [15] and conducted related experiments to verify the usability of the method. In addition, Ref. [15] introduced an operator matrix to correlate various indicators and finally obtained a variable or indicator to express all aspects of the network. Since not all mutation phenomena originate from network anomalies, this may lead to model misjudgment. To cope with the new requirements of network security brought by the complexity of cellular networks, Ref. [16] proposed an anomaly detection algorithm based on Bayesian decision rules and applied it to mobile user profiles to verify the method’s feasibility. In addition, the algorithm specializes in privacy protection, however, the algorithm’s analysis function also violates users’ privacy. Ref. [17] proposed a multi-level hierarchical Kohonen network (K-Map) for intrusion detection, where each layer of the hierarchical graph is modeled as a simple winner-takes-all K-Map. This multi-level hierarchical K-Map structure has the advantage of low computational complexity, avoids costly peer-to-peer computations by organizing the data into clusters, and reduces the size of the network. Ref. [18] proposed an anomaly detection algorithm based on an unrestricted α-stable first-order model and statistical hypothesis validation by automatically selecting a flow window to be used as a reference, compared with an observed flow window. The algorithm of [18] focuses on detecting two anomaly types: floods and flash crowds. Ref. [19] proposed a flow-based aggregation technique (FSAS), which greatly reduces the amount of monitored data and handles large amounts of statistical and grouped data. A stream, or IP stream, is a given series of IP packets. The FSAS set flow-based statistical feature vectors reports to the acute neural network classification model. Ref. [20] developed a new statistical decision-theoretic framework for network traffic using Markov chain modeling. The algorithm first formulates the optimal anomaly detection problem for the composite model of [20] via generalized likelihood ratio check (GLRT). However, this algorithm leads to a very expensive combinatorial optimization problem. Then, Ref. [20] developed two low-complexity anomaly detection algorithms, the cross-entropy-based and GLRT-based methods. Ref. [21] implemented an algorithm developed in the SRI-based NIDES (next-generation intrusion detection expert system) project. In addition, Ref. [21] also developed three OSPF routing protocol insider attacks to evaluate the effectiveness of detection capabilities. Ref. [22] developed an anomaly detection method for large networks. The algorithm first uses a Kalman filter to filter out “normal” traffic and judges by comparing the predicted traffic matrix with the actual traffic matrix. Then, detect whether there is any abnormality in the remaining filtering process. Ref. [22] here explains how any anomaly detection method can be viewed as a problem in statistical hypothesis testing. Ref. [23] believed that if the joint distribution of multi-dimensional data can be effectively expressed, one can try to estimate the tail probability of each point, and then the abnormal situation can be well evaluated. To date, Ref. [23] proposed a copula-based anomaly detection algorithm. The copula is a statistical probability function for efficiently modeling dependencies among multiple random variables. Ref. [23] used a non-parametric method, obtains empirical copula through empirical cumulative distribution (Empirical CDF), and then estimates the tail probability of joint distribution of all dimensions through empirical copula. The advantage of the algorithm based on data feature statistics is that it is simple to implement and can also identify abnormal traffic in unknown networks, but it is prone to misjudgment.

Algorithms based on machine learning have a strong learning ability and can effectively deal with unknown abnormal network traffic. Ref. [24] proposed a 5G-oriented network defense architecture. To this end, Ref. [24] used deep learning techniques to analyze network traffic by extracting features from it. In addition, the architecture allows automatic the adjustment of the configuration of the network fabric to manage traffic fluctuations. Experiments show that the method can adaptively adjust the anomaly detection system and optimize resource consumption. Ref. [25] proposed an anomaly-based NIDS implemented using deep learning techniques. The method demonstrates the ability and adaptability to infer partial knowledge from incomplete data. With the advent of the Internet of Things, the need to process streaming data in real-time has become critical. To this end, Ref. [26] proposed a hybrid data processing model using GWO and CNN for anomaly detection. In order to increase the learning ability of the model, both GWO and CNN learning methods have been enhanced; for the first, the generative ability to explore, exploit and initialize the population is improved; for the second, the dropout function is improved. Ref. [26] first used GWO for feature selection to obtain the best trade-off between two objectives, i.e., reducing the error rate and minimizing the feature set. Then, Ref. [26] used CNN for network anomaly classification. In Ref. [27], in order to deal with the security issues brought by SDN, a deep neural network model was constructed and trained using the NSL-KDD dataset. During the training process, only six basic features among the forty-one features were selected for training. However, the dataset selected by this method is not in the SDN environment and is not necessarily suitable for the SDN environment. To improve the reliability of SDN, Ref. [28] proposed a hybrid deep learning-based anomaly detection scheme for suspicious flow detection in social multimedia environments. The scheme consists of an anomaly detection module and an end-to-end data transfer module. The anomaly detection module utilizes a modified, restricted Boltzmann machine and a gradient descent-based support vector machine to detect anomalous traffic. The end-to-end data transmission module is designed to meet the strict QoS requirements of SDN. Since existing anomaly detection solutions all require a large number of datasets for offline training, Ref. [29] proposed a neural network-based anomaly detection system with dynamically updatable training models—Griffin, which utilizes an ensemble of autoencoders from normal and abnormal traffic which is jointly screened in the traffic, and the autoencoder, which is updated according to the mean square error. In order to solve the problem of high memory consumption, low accuracy, and the high processing and overhead of detection methods in the IoT environment, Ref. [30] proposed sFlow and sampling based on adaptive rotation training, combined with the Snort intrusion detection system and deep learning-based model, which is helpful for the IoT in cases of various types of DDoS attacks. Due to the decoupled nature of SDN, Ref. [30] obtained the required parameters by programming network devices. First, in the data plane, in order to reduce the switches’ processing and network overhead, Ref. [30] distributed the deployment of sFlow and sampling based on the adaptive round-robin. Second, to optimize the detection accuracy in the control plane, Ref. [30] deployed the Snort IDS with the SAE deep learning model. Algorithms based on machine learning have a stronger learning ability and can learn from incomplete traffic characteristics to abnormal traffic, but the models generally have high computational complexity and are not suitable for high-response environments.

## 3. Materials and Methods

TSMASAM is a deep learning model based on ensemble learning and the self-attention mechanism. Combining the self-attention mechanism and ensemble learning makes up for the relationship between data that cannot be learned by ensemble learning. The module frame is shown in Figure 2. TSMASAM consists of two parts: sample association learning and integrated detection network. The dataset we adopted (InSDN dataset) was generated by simulating the environment under four virtual machines; the first virtual machine was Kali Linux, which represents the attacker server; the second virtual machine was Ubuntu 16.4, which was used as ONOS the controller; the third was the Ubuntu 16.4 machine as a Mininet and OVS switch; the fourth virtual machine was a Metasploitable 2-based Linux machine as an exploit service to demonstrate the exploit. The controller of SDN was implemented using the open source tool ONOS. This dataset contains anomalous traffic from inside and outside attackers targeting the controller. Our approach builds on this to detect traffic data passing through an SDN controller.

### 3.1. Data Preprocessing

The experimental flow is shown in Figure 3. Before the data enter the model, the data are preprocessed. This paper first normalizes the data. If there is missing information in the dataset, the data row is deleted. Finally, this paper uses the hierarchical leave-out method to split the dataset into two parts: the training set and the test set. Due to the unbalanced nature of the data, this paper performs oversampling operations on the training set.

### 3.2. Related Definitions

**Definition** **1.**
*The dataset is represented as $X=\left[x_{1},\ldots ,\;x_{N}\right]\in R^{m\times N}$, where each sample is represented as $x_{i}=\left[x_{i}^{1},\ldots ,x_{i}^{m}\right]\in R^{m}$, by m-dimensional feature composition.*


**Definition** **2.**
*$F=\left[{f\left(X\right)}_{1},\ldots ,{f\left(X\right)}_{k}\right]$ is expressed as the set of base learners with the number of base learners k.*


**Definition** **3.**
*In the abnormal traffic detection model, input data $X=\left[x_{1},\ldots ,\;x_{N}\right]$ is given. The model aimed to learn a function H(X) to classify samples. Finally, according to the classification result, determine whether the sample $x_{i}$ is abnormal:*

(1)
$$\begin{array}{c} \hat{y_{i}}=\left\{\begin{array}{c} 1\;,if\;H\left(x_{i}\right)=1\\ 0\;,if\;H\left(x_{i}\right)=0 \end{array}\right. \end{array}$$

*wherein*

$\hat{y_{i}}=1$

*means that the function H(X) predicts that the data sample*

$x_{i}$

*is abnormal, and*

$\hat{y_{i}}=0$

*implies that the function H(X) predicts that the data point*

$x_{i}$

*is normal.*


### 3.3. Sample Associative Learning

To explore the correlation between multiple sample features, the model introduces a self-attention mechanism to automatically learn the correlation between samples, captures the feature space’s internal structure, and uses a convolutional network to construct a sample embedding to express the relationship between data samples. Given a set of input samples $X=\left[x_{1},\ldots ,\;x_{N}\right]\in R^{m\times N}$, the self-attention mechanism is used to learn the sample $x_{i}=\left[x_{i}^{1},\ldots ,x_{i}^{m}\right]\in R^{m}$ relationship between traffic characteristics. Self-attention maps samples to three different feature spaces, resulting in three vectors (query vector $q_{i}\in R^{D_{k}}$, key vector $k_{i}\in R^{D_{k}}$, and value vector $v_{i}\in R^{D_{v}}$):(2)$$\begin{array}{c} Q=W_{q}X{\in R}^{D_{k}\times N} \end{array}$$(3)$$\begin{array}{c} K=W_{k}X{\in R}^{D_{k}\times N} \end{array}$$(4)$$\begin{array}{c} V=W_{v}X{\in R}^{D_{v}\times N} \end{array}$$
where $W_{q}{\in R}^{D_{k}\times D_{x}}$, $W_{k}{\in R}^{D_{k}\times D_{x}}$, $W_{v}{\in R}^{D_{v}\times D_{x}}$ are the parameter matrix of feature mapping, and the $Q=\left[q_{1},\ldots ,q_{N}\right]$, $K=\left[k_{1},\ldots ,k_{N}\right]$, $V=\left[v_{1},\ldots ,v_{N}\right]$ matrix consists of the query vector, key vector and value vector, respectively. The purpose of setting *Q*, *K*, and *V* is to find the correlation coefficient with other features by calculation, calculate a weight for each feature, and then obtain a weighted result to judge the relationship between each feature and other features. At most, the detection efficiency of traffic is improved by learning the information in these attention values. The main work of self-attention is to calculate the dot product of the query *Q* and all *K*, scale it, derive the weight of the value *V* through the softmax function, and then multiply the value *V* and the weight to obtain the attention value:
(5)μn=att((K,V),qn)(6)=∑j=1Nanjvj(7)=∑j=1Nsoftmax(s(kj,qn))vj(8)=∑j=1Nexp(s(kj,qn))∑zexp(s(kz,qn))
among them, j,n∈[1,N] is the position of the input vector and the output vector sequence; $a_{nj}$ represents the weight value of the nth output concerned with the jth input; $q_{n}$ represents the query vector of the nth input sample; $k_{j}$ represents the key of the jth input; $v_{j}$ represents the value of the jth input, which contains the input information. Therefore, the calculated attention value is equivalent to the attention value between the ith sample and the 1st, 2nd, and *i*th inputs, that is, the correlation between each input. After obtaining the calculated attention value $\;\mu_{i}$, map $\mu_{i}$ to a new feature space to obtain the embedding vector $e_{i}\in R^{D_{s}\times k}$:(9)$$\begin{array}{c} e_{i}=softmax\left(\bar{f}\left(\mu_{i}\right)\right) \end{array}$$
where $\overset{-}{f}\left(\mu_{i}\right)$ is the feature mapping function. After the above calculation, the embedding vector $e_{i}$ is finally obtained, and $E=\left[e_{0},\ldots ,e_{N}\right]\in R^{D_{s}\times k\times N}$ is the sample embedding matrix composed of the embedding vectors. In this paper, the mapping function we choose is the convolutional neural network (CNN).

### 3.4. Ensemble Detection Network

#### 3.4.1. Auto Encoder

We use an autoencoder to denoise the original to obtain representative feature information in the samples. After processing by the autoencoder, we obtain the reconstructed information vector $\overset{-}{X}=\left[{\overset{-}{x}}_{1},\ldots ,{\overset{-}{x}}_{N}\right]\in R^{m\times N}$:(10)$$\begin{array}{c} h_{i}=g\theta_{1}\left(x_{i}\right)=\sigma \left(W_{1}\times x_{i}+b_{1}\right) \end{array}$$(11)$$\begin{array}{c} \bar{x_{i}}=g\theta_{2}\left(x_{i}\right)=\sigma \left(W_{2}\times x_{i}+h_{2}\right) \end{array}$$
where $h_{i}$ is the latent feature learned by the auto-encoder from the input information $x_{i}$, and $g\theta_{1}\left(x_{i}\right)$ and $g\theta_{2}\left(x_{i}\right)$ are the encoder and decoder functions in the auto-encoder. The encoder function $g\theta_{1}\left(x_{i}\right)$ and the decoder function $g\theta_{2}\left(x_{i}\right)$ are composed of a multi-layer fully connected network for feature transformation. The purpose of the encoder is to perform feature transformation on the sample features, and the purpose of the decoder is to reconstruct the original data from the latent features $h_{i}$ obtained by the encoder to obtain the decoded data $\overset{-}{x_{i}}$.

#### 3.4.2. Stacking Ensemble Detection Network

The stacking ensemble detection network learns to train the base learner by reconstructing the data $\overset{-}{X}$. In this paper, CNN, LSTM, and LENET networks are used as the heterogeneous base learner of the network, and the classification judgment of traffic is made according to the reconstructed data $\overset{-}{X}$:(12)$$\begin{array}{cc} \delta_{i} & ={f\left(\overset{-}{X}\right)}_{i}\in R^{N} \end{array}$$(13)$$\begin{array}{cc} \Delta & =F\left(\overset{-}{X}\right)\in R^{D_{o}\times k\times N} \end{array}$$
where ${f\left(\overset{-}{X}\right)}_{i}$ is the base learner, $\delta_{i}$ is the outlier matrix obtained after the base learner ${f\left(\overset{-}{X}\right)}_{i}$ detection, and $\Delta =\left[\delta_{0},\ldots ,\delta_{k}\right]\in R^{D_{o}\times k\times N}$ is composed of outliers obtained by the base learner. The learner can be any supervised classifier. In theory, heterogeneous base learners can make more robust coarse-grained detection. Diversity and heterogeneity among base learners can provide different perspectives for classification.

In the second layer of the stacking ensemble detection network, we designed a fusion module to train the fusion module by using the long-term dependencies between samples captured by the self-attention mechanism and the prediction results of the base learner as a new dataset. Based on the embedded vector $c_{i}$ and the outlier matrix $\delta_{i}$, the stacking ensemble detection network performs a dot product operation on the embedded vector $c_{i}$ and the outlier matrix $\delta_{i}$ through a fusion module then obtains the final detection result through a fully connected layer:
$$\begin{array}{cc} o_{i} & =softmax\left(\sum_{j=0}^{k}CF\left(c_{ij},\delta_{ij}\right)\right)\\ & =softmax\left(\sum_{j=0}^{k}c_{ij}\delta_{i}j\right) \end{array}$$
where $CF\left(c_{ij},\delta_{ij}\right)$ is the embedding fusion function, and $o_{i}$ is the outlier vector obtained by the fusion function. We choose the dot product method to fuse the embedding vectors in our method. In theory, an excellent fusion method can effectively exploit the information in the embedding vector.

We design a new loss function to fully consider the sharing of each base model to the model as a whole. The loss function of our method is calculated based on the cross-entropy loss function:$$\begin{array}{c} L\left(\hat{Y},Y,\Delta \right)=W_{0}\times CE\left(\hat{Y},Y\right)+\sum_{t=1}^{k}W_{t}\times CE\left(\delta_{t},Y\right)+log\left(1+\prod_{t=1}^{k}CE\left(\delta_{t},Y\right)\right) \end{array}$$
where CE is the cross-entropy loss function, $W_{t}$ is the weight value with a sum of 1, *Y* is the label set, $\hat{Y}$ is the outlier prediction made by the method in this paper, and Δ is the outlier prediction value set made by the base learner set. $W_{0}\times CE\left(\hat{Y},Y\right)$ is the cross entropy between the predicted value made by the method in this paper and the label set *Y*. $W_{t}\times CE\left(\delta_{t},Y\right)$ is the cross entropy between the predicted value of each base learner and the label set Y, which aims to fully consider the pre-detection results of the base learner during the training process. $log\left(1+\prod_{t=1}^{k}CE\left(\delta_{t},Y\right)\right)$ is used as the regular term of the loss function to prevent the model from falling into an overfitting state. $1+\prod_{t=1}^{k}CE\left(\delta_{t},Y\right)$ guarantees that the function value is always greater than 1. $W_{t}$ is used as a hyperparameter in the model training process to adjust the weight of the different items.

## 4. Experiment and Analysis

### 4.1. Experimental Environment and Datasets

The TSMASAM designed in this paper is implemented based on Python3, Pytorch1.2 and Numpy. The four CPU models are Intel(R), Xeon(R), CPU E5-2620 v2 @ and 2.10GHz, the graphics card model is Matrox G200eR2, and the PyTorch version is the 1.2.0 server environment. The InSDN dataset is generated from environment simulations under four virtual machines; the first virtual machine is Kali Linux, which represents the attacker server; the second virtual machine is Ubuntu 16.4, which acts as the ONOS controller; the third Ubuntu 16.4 machine, as a Mininet and OVS switch; the fourth virtual machine is a Metasploitable 2-based Linux machine that serves as an exploit service to demonstrate the exploit. The controller of SDN is implemented using the open source tool ONOS. This dataset contains anomalous traffic from inside and outside attackers targeting the controller.

This paper evaluates the performance of TSMASAM through two experiments. This paper applies the network traffic simulation dataset [31] under the SDN architecture. It is derived from the SDN virtual environment and is constructed by multiple virtual machines using the SDN network architecture. Since the abnormal flow in [31] is much more than normal, the research team randomly selects the flow data to simulate the sample imbalance phenomenon in the real environment. The processed dataset contains 76,825 network traffic, of which 8401 are abnormal, accounting for approximately 10.94% of the overall sample. The research team used the hierarchical set-out method to divide the dataset into the training set and test set, each of which contains a total of 84 traffic eigenvalues. This dataset includes seven network attack types (Probe, DDoS, DoS, BFA, Web-Attack, BOTNET, and U2R). The distribution of each network attack is shown in Figure 4. In this experiment, seven network attack types (Probe, DDoS, DoS, BFA, Web-Attack, BOTNET, and U2R) are regarded as abnormal, and the rest are regarded as normal traffic data.

To verify the generalization ability, not only the InSDN dataset [31] but also the KDD99 and UNSW-NB15 datasets are used in the experiments.

### 4.2. Evaluation Indicators

There are only two types of anomaly detection targets in this paper. The positive examples are normal data, and the negative ones are abnormal. The classification results of the experiments can be divided into the following four categories:True positives (TP): TP represents the proportion of abnormal behavior correctly identified as abnormal behavior;False positives (FP): FP represents the proportion of normal behavior incorrectly identified as abnormal behavior;False negatives (FN): FN represents the proportion of abnormal behavior incorrectly identified as normal behavior;True negatives (TN): TN represents the proportion of normal behavior correctly identified as normal behavior;

After classifying the results, this paper evaluates the algorithm’s performance by Precision, Recall, and F1-score.
(16)Precision=TPTP+FP;(17)Recall=TPTP+FN;(18)F1-score=2×Precision×RecallPrecision+Recall
among them, precision indicates the rate of correct identification of abnormal behavior and normal behavior. Recall describes how many real positive examples in the test set are selected by the binary classifier. The core idea of the F1-score is that, while improving precision and recall as much as possible, we also want the difference between the two to be as small as possible.

### 4.3. Performance Testing and Analysis

#### 4.3.1. Performance Testing

The experimental results of detection performance are shown in Table 1 and Table 2, and the model loss curve is shown in Figure 5.

The abscissa represents the epoch number (a total of 200 epochs), and the ordinate represents the loss value of the model. It can be seen from the figure that the loss value of TSMASAM shows a small range of fluctuation in the early training period, and then quickly converges and maintains a low oven for a long time.

The experimental results are shown in Table 1 and Table 2. It can be seen from the table that TSMASAM has a precision of 99.72%, a recall of 99.96%, and an F1-score of 99.84%. (1) Comparing machine learning methods (COPOD, HBOS, IForest, VAE, ECOD, and LOF): on the [31] dataset, the machine learning method can achieve the highest accuracy of 82.19%; the machine learning algorithm can achieve the highest recall rate of 76.30%; the machine learning algorithm can achieve the highest F1 score of 77.92%. (2) Comparing the ensemble learning methods (XGBOD, LSCP, SUOD, and LODA): on the [31] dataset, the highest performance can reach 99.98%. The detection performance of the method proposed in this paper is better than most of the comparison algorithms but weaker than the XGBOD method.

The experimental results of detection performance are analyzed as follows: the traditional machine learning algorithm has limited ability and poor generalization ability, which makes the model’s learning of traffic characteristics too limited. The learning of datasets with long-term associations is not sufficient, so the performance is lower.

Due to the large randomness of traffic, it is difficult to learn suitable feature information. The detection method based on ensemble learning ensures the diversity of weak classifiers and fully considers each base model in decision making so that the results obtained are better than traditional ones. The machine learning method works well.

The method proposed in this paper learns the long-term dependencies between data samples through the self-attention mechanism and convolutional network and transfers them to the ensemble learning model in the form of sample embedding so that the ensemble learning model can more accurately model the process. Therefore, the detection mechanism proposed in this paper is more stable, and the detection effect is not weaker than other integrated learning methods.

#### 4.3.2. Control Group Experiment

Table 3 shows the performance impact of the sample associative learning on TSMASAM. The precision, recall and F1-score of stacking are 0.8059, 0.8893, and 0.8390. The sample associative learning improves TSMASAM by an average of 15.37% on each index.

Table 4 shows the performance of the selected base learners (CNN, LSTM, and LENET) in the ensemble learning model on the [14] dataset. It can be seen from the table that LSTM and LENET have better performance in terms of recall rate, while CNN achieves 99.73% in precision, but the recall rate is 0.34% lower than that of LSTM and LENET. On the other hand, our method achieves an average improvement of 7.28% in F1-score over base learners (CNN, LSTM, and LENET).

Table 5 shows the performance of the proposed method on different datasets. Under the KDD99 dataset, our method achieves 99.78% precision, 99.81% recall, and 99.78% F1-score. On the UNSW-NB15 dataset, our method achieves 80.51% precision, 92.93% recall and 86.27% F1-score.

Table 6 shows the impact of different base learners on the detection results of our method. In addition, we also show the average computation time of TSMASAM for each traffic datum in the table. From it, we can see that our method achieves the best performance when the Kernel_size of CNN is 5, the number of hidden layers of LSTM is 3, and the number of hidden layers is 128.

The analysis of the control group experiment is as follows: introducing a self-attention mechanism in ensemble learning to capture long-term dependencies between data samples can improve the model’s detection performance. After adding the self-attention mechanism, the method in this paper improves the original detection index by 15.37% on average. In the experiments of base learners, our method integrates the detection capabilities of each base learner well. In experiments on different datasets, our method shows good generalization ability and performs well on KDD99 and UNSW-NB15. Since the method in this paper seeks to improve the detection performance, such as introducing a self-attention mechanism, the computational complexity is increased. Therefore, it is not suitable for application scenarios with high response speeds.

## 5. Conclusions

While SDN technology brings a certain degree of convenience to people, it also brings security risks due to its own design. In order to protect the security of supporting SDN technology, this paper proposes an intrusion detection algorithm, TSMASAM, based on ensemble learning. TSMASAM introduces a self-attention mechanism to capture the correlation between data features to improve the integration effect of the model; TSMASAM achieves the detection and identification of abnormal network traffic by integrating the sample embedding obtained above and the inspection results of the heterogeneous base learner. The purpose is to effectively improve the effect of an integrated detection of abnormal traffic in industrial scenarios using SDN technology.

The dataset used in this paper is generated by simulating the SDN network built in the virtual machine environment, and the traffic data of the controller is collected. Therefore, the model in this paper is mainly oriented to the abnormal traffic detection of the controller in the SDN network. The model proposed in this paper increases the training time and running time to a certain extent in order to consider the influence of the base learner on the model. In the real environment, the intrusion detection algorithm also pays attention to timeliness, so the next step is to study how to maintain the algorithm’s performance while shortening the running time.

## Figures and Tables

**Figure 1 sensors-22-07436-f001:**
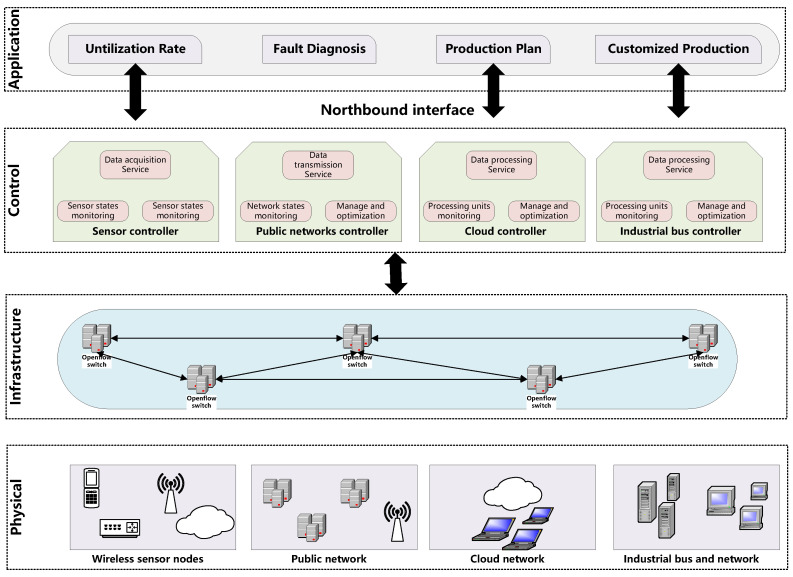
SDN network architecture diagram.

**Figure 2 sensors-22-07436-f002:**
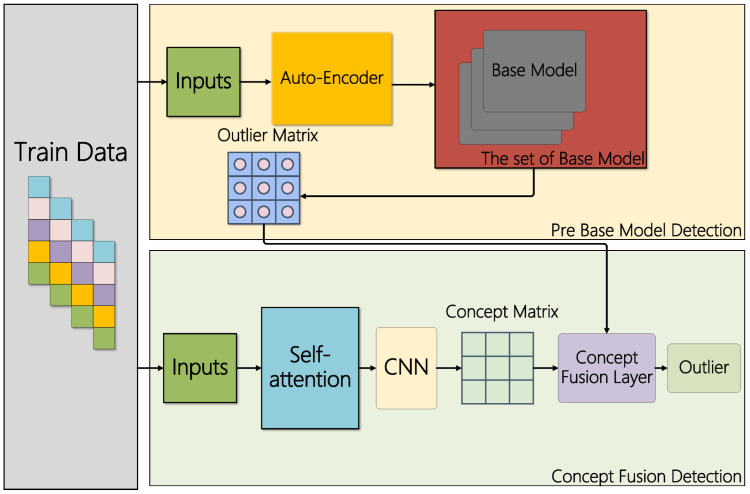
Adaptive ensemble learning model based on the self-attention mechanism.

**Figure 3 sensors-22-07436-f003:**
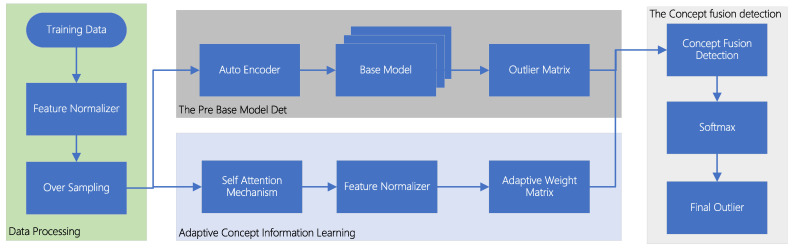
Flow chart of TSMASAM experiment.

**Figure 4 sensors-22-07436-f004:**
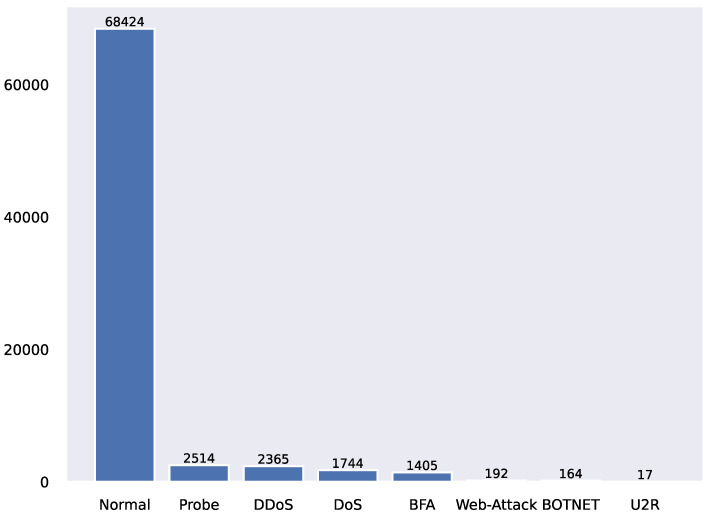
Label distribution in the dataset.

**Figure 5 sensors-22-07436-f005:**
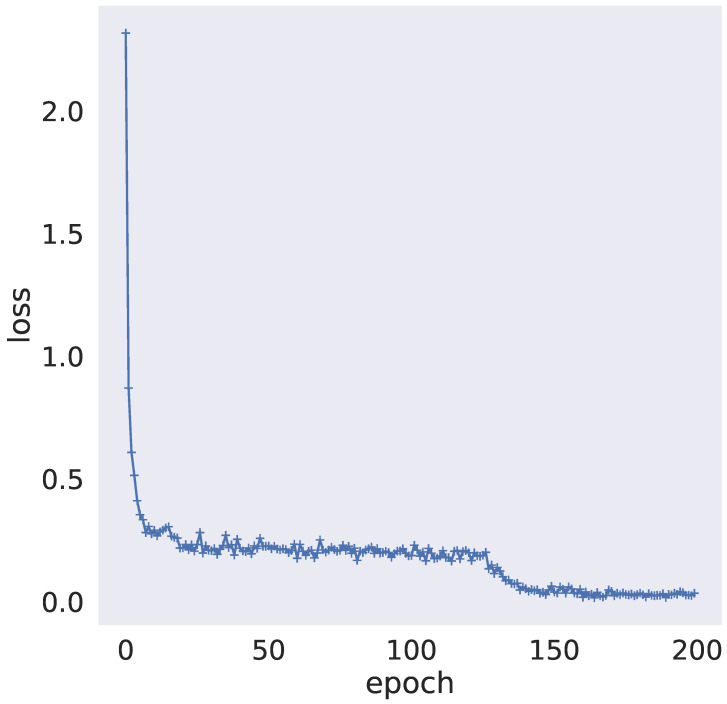
The loss curve of the TSMASAM model.

**Table 1 sensors-22-07436-t001:** Comparative experiment between TSMASAM and the machine learning-based anomaly detection algorithm.

Model	Precision	Recall	F1-Score
**COPOD**	0.7799	0.7057	0.7403
**HBOS**	0.7968	0.7630	0.7792
**IForest**	0.8108	0.7043	0.7491
**VAE**	0.7988	0.6912	0.7378
**ECOD**	0.8219	0.6757	0.7323
**LOF**	0.8002	0.5627	0.6456
**TSMASAM**	**0.9972**	**0.9996**	**9984**

**Table 2 sensors-22-07436-t002:** Comparative experiment between TSMASAM and the anomaly detection algorithm based on ensemble learning.

Model	Precision	Recall	F1-Score
**XGBOD**	**0.9998**	**0.9998**	**0.9998**
**LSCP**	0.7784	0.6240	0.6895
**SUOD**	0.7784	0.7920	0.7885
**LODA**	0.7656	0.6278	0.6892
**TSMASAM**	0.9972	0.9996	9984

**Table 3 sensors-22-07436-t003:** The impact of the Sample Associative Learning on the model.

Model	Precision	Recall	F1-Score
**Stacking**	0.8059	0.8893	0.8390
**TSMASAM**	**0.9972**	**0.9996**	**9984**

**Table 4 sensors-22-07436-t004:** The performance of the base learner on the dataset.

Model	Precision	Recall	F1-Score
**CNN**	0.9973	0.9966	0.9970
**LSTM**	0.8017	1.0000	0.8899
**LENET**	0.8018	1.0000	0.8900

**Table 5 sensors-22-07436-t005:** TSMASAM performance on other datasets.

Model	Precision	Recall	F1-Score
**KDD99**	0.9978	0.9981	0.9978
**UNSW-NB15**	0.8051	0.9293	0.8627
**InSDN**	0.9972	0.9996	0.9984

**Table 6 sensors-22-07436-t006:** Ablation experiment.

Base Leaner	Precision	Recall	F1-Score	Time
CNN(kernel_size = 5), Lenet, Lstm(hidden_size = 128, hidden_layer = 3)	0.9978	0.9981	0.9978	1.3970-10
CNN(kernel_size = 3), Lenet, Lstm(hidden_size = 128, hidden_layer = 3)	0.9967	0.9971	0.9969	8.2888-10
CNN(kernel_size = 5), Lenet, Lstm(hidden_size = 128, hidden_layer = 10)	0.8016	1.0000	0.8899	8.9873-10
CNN(kernel_size = 5), Lenet, Lstm(hidden_size = 64, hidden_layer = 3)	0.9942	0.9899	0.9920	8.3121-10
CNN(kernel_size = 3), Lenet, Lstm(hidden_size = 64, hidden_layer = 3)	0.9899	0.9994	0.9946	8.8011-10
CNN(kernel_size = 5), Lenet, Lstm(hidden_size = 64, hidden_layer = 10)	0.8016	1.0000	0.8899	8.8915-10

## Data Availability

Restrictions apply to the availability of these data. Data were obtained from UCD ASEADOS Lab and are available at https://aseados.ucd.ie/?p=177 with the permission of UCD ASEADOS.

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
