# Peer review of "An Anomaly Detection Algorithm Based on Ensemble Learning for 5G Environment"

_sensors, 2022, doi:10.3390/s22197436_

Round 1

Reviewer 1 Report

The authors present a strategy to detect anomaly traffic detection in wireless networks. 

In its current form, the manuscript cannot be accepted. It is not specified how the proposal contributes to improving the shortcomings that current methodologies have nor the advances that are achieved.

On the other hand, the obtained results are not adequately presented, since a basic comparison with similar methodologies is missing. 

Extensive editions and proofreading are required.

Author Response

Dear Reviewer,

Thank you very much for your sincere suggestions for our manuscript. The detailed response is in the attachment.

Best regards,

Reviewer 2 Report

After having assessed the suitability for publication of the Manuscript ID: sensors-1905353, having the title "An anomaly detection algorithm based on ensemble learning for 5G environment", I have distinguished several elements that from my point of view should be made less confused and more comprehensible by the authors in view of improving the quality of the manuscript. Therefore, I have devised and wrote a series of comments to the authors of the manuscript under review.

In this paper, the authors develop a deep learning model for anomalous traffic identification based on ensemble learning in the case of 5G environment. The Manuscript ID: sensors-1905353 is interesting. However, the article under review can be improved if the authors address the following aspects in the text of the manuscript and reflect them clearly point-by-point within the cover letter:

1. Aspects regarding the impact of the latency and of the network's size that generates the traffic on the proposed approach. Considering that the equipment within a Software Defined Network (SDN), in a 5G network, must be able to properly handle a plethora of devices at any given time, it is of extreme importance to have a clear understanding of how these devices will interact with the network and what kind of impact they will have. Considering this aspect, the speed of interaction between devices and the network will be influenced by the number of virtualized resources. This means that if more speed is needed, more virtualized resources can be introduced. However, virtualizing resources could also result in a significant amount of latency. Therefore, it is important to have a clear strategy that weights the advantages and disadvantages of virtualization before making any decisions. The manuscript will benefit if the authors provide a clear insight within the manuscript through which they strengthen how did their proposed approach take into consideration these important aspects regarding the latency and the number of virtualized resources. How were these aspects tackled by the authors when devising their proposed approach and what particular experiments concerning these aspects have been conducted in order to assess the impact that the latency and the network's size that generates the traffic that must be analyzed have on the proposed approach?

2. Aspects regarding the proper planning and carrying out of the maintenance process within a Software Defined Network (SDN). The maintenance process within a 5G network is critical for supporting the high speeds and large data requirements of the emerging applications and services. However, a Software Defined Network (SDN) can make it difficult to manage the underlying devices, especially when expanding the network. SDNs fall short in this area, making it difficult to manage actual devices, especially when upscaling the network. This is a significant drawback of SDN that must be considered. How did the authors take into account this aspect when devising their proposed approach? What particular experiments have been conducted in order to assess the added impact of the retraining process of their proposed approach over the maintenance operations?

3. Aspects regarding the comparison to the level of security offered by robust conventional security standards strengthened over the decades. Considering that Software Defined Networks (SDNs) do not make use of conventional routers and switches, the security typically included with those devices is omitted. For instance, the security that conventional firewalls provide is not present in an SDN. This makes SDNs more vulnerable to external threats in comparison to the robust security offered by conventional equipment, security that has been strengthened over the years. How does the proposed approach of the authors compare to the level of security offered by robust security standards strengthened over the course of decades in conventional approaches, security that is offered by conventional equipment that does not exist within SDNs?

4. Lines 19-183, the "Introduction" and "Related Works" sections – the state of knowledge. In the current form of the manuscript, the authors have performed within these two sections a survey of what has been done up to this point in the scientific literature. I do not contradict the value of these papers, or their relevance in this context, but I consider that the article under review will benefit if the authors carefully rewrite these sections. The authors should perform the literature review in a manner that emphasizes for the main scientific works that have been cited in the text of the manuscript the contribution that was made to the existing state of art, the approach that was employed along with a short description of the most important results that have been obtained along with the existing unsolved issues of the referenced studies. By doing so, the authors will be able to encompass their study in a broad body of knowledge and highlight unsolved problems that still exist regarding the subject that their manuscript addresses. The manuscript will benefit from including a more detailed review of the literature on 5G network slicing and especially on standardization activities related to anomaly detection algorithms. Currently, the authors only provide a limited review of these topics, which is not sufficient to support the importance of the addressed gap. Furthermore, the lack of proposing clear directions for standardization activities that standardizing committees of international bodies should pursue related to 5G network slicing and anomaly detection traffic algorithms make it difficult to assess the real potential of the proposed method for being used in a real-working production environment, namely a real 5G network, operating 24 hours a day, 7 days a week.

5. Aspects regarding the relevance of the conducted experiments and obtained results. As the experimental tests have not been conducted on data collected from a real-working production environment, namely a real operating 5G network, operating 24 hours a day, 7 days a week, the results of these experiments cannot accurately simulate or detect the potential abnormalities that may occur in an actual network setting. Furthermore, the tests seem to be geared towards detecting specific types of anomalies or malicious traffic, rather than providing a general way to flag any type of strange or unexpected behaviorThis could lead to false positives or false negatives in a real-world application scenario. Therefore, while the experiments may provide some valuable insights, they are not entirely relevant to the task of detecting anomalies in a real-world 5G network. The results of the experimental tests cannot be accurately reproduced in a production environment. The 5G network is a lot more complex and produces huge data streams than the network traffic dataset used in the experimental tests and, as such, the results cannot be extrapolated to reflect the performance of the model in a real-world 5G network. This aspect should be acknowledged within the limitations of the proposed approach.

6. Lines 1-17, the "Abstract" of the paper. The manuscript will benefit if in its "Abstract", along with the elements already presented, the authors also declare and briefly justify the novelty of their work. In addition to this, the authors should briefly present the most important obtained numerical result(s).

7. The "Introduction" section - the gap in the current state of knowledge. At Lines 90-92, the authors state: "In order to effectively improve the effect of abnormal traffic detection, we propose an abnormal traffic detection method based on the stacking method and self-attention mechanism(TSMASAM)." Firstly, it is not clear why should be improved "the effect of abnormal traffic detection" instead of the "abnormal traffic detection" itself. Secondly, I consider that in the Manuscript ID: sensors-1905353, the gap in the current state of knowledge is not enough highlighted. After having performed the literature survey in an appropriate manner, the authors will be able to pinpoint clearer an exact deficiency, an unsolved problem, a gap that still exists in the current body of knowledge that needs to be filled, therefore justifying the need and novelty of their study. Otherwise, without identifying and stating clearly this gap, the study from the manuscript under review does not justify its need, importance and novelty. Emphasizing the gap will improve the manuscript under review on multiple plans, as the identified deficiency, the identified unsolved problem will offer the authors great opportunities to highlight and prove, when discussing their results, the contribution, the advancement that the conducted research has brought to the existing state of knowledge.

8. The "Introduction" section – the novel aspects of the conducted study. After properly highlighting the gap in the current state of knowledge, it will benefit to state the novel aspects of the conducted study and afterwards, at the end of the "Introduction" section, the authors should present the structure of their paper, under the form: "The rest of the paper is structured as follows: Section 2 contains…".

9. The "Materials and Methods" section is missing, being partially replaced by a "Method" section. I consider that the authors must devise a proper "Materials and Methods" section. In order to bring a benefit to the manuscript, the authors should mention early in the "Materials and Methods" section, the choices they have made in their study. The authors should state what has justified using the given method, what is special, unexpected, or different in their approach. If the authors make use of a standard or usual procedure, this aspect should also be mentioned upfront, from the very beginning. I consider that the manuscript under review will benefit if the authors make all of these aspects as clear as possible to the readers starting from the first sentence of the paragraph in order to give them a clear idea of what the entire paragraph is about.

10. The "Materials and Methods" section - the software and the detailed hardware configuration. It will benefit the paper if, along with the elements already presented, the authors provide specific details regarding the hardware and software configurations of the platform used to obtain the results.

11. Issues regarding the tables and figures within the paper. In its actual form, the manuscript contains a lot of insufficiently interpreted and explained tables and figures (for example, Figures 1 and 2 are not interpreted or discussed, Figure 3 is not even cited, Figure 4 is improper cited as "Fig. 4" instead of "Figure 4"). The authors must explain and analyze in detail all the tables and figures that have been inserted within the manuscript, it is not suitable to put the reader in the situation of interpreting, analyzing, continuing or refining the study from the manuscript under review.

12. The equations within the manuscript. The equations within the manuscript should be explained, demonstrated or cited, as there are some equations that have not been introduced in the literature for the first time by the authors and that are not cited (for example equations (16)-(18)).

13. The mathematical formalism. The authors must improve the mathematical formalism and pay more attention to the details.  For example, the notation oi used within the equation (1) should be defined, explained. What is the meaning of this notation?

14. The dataset – the data division ratioThe authors state at Lines 231-232:"The research team used 70% of the processed data set as the training set, and each piece of data contained a total of 84 traffic feature values. The authors should explain in the paper the reasons for choosing this data division ratio. The paper will benefit if the authors present more details regarding the results obtained during various tests, for all the different tested ratio values, up to the moment when the chosen ratio has proven to be the best (or suitable) approach and what was the criterion/performance metric used in choosing this ratio.

15. Details regarding the missing data or the abnormal values. The manuscript will benefit if the authors provide in the paper more details regarding the way in which they intend to solve the problems related to missing data or abnormal values (caused by measurement errors), as these are highly likely to occur in a real production environment, namely a 5G network.

16. The generalization capability of the developed approach. Can the authors mention how much of their model is being influenced by the used data or to which extent the model can be easily applied to other situations, when the datasets are different? In this way, the authors could highlight more the generalization capability of their approach in order to be able to justify a wider contribution that has been brought to the current state of art.

17. The "Experiment and analysis" section. In this section the authors should highlight current limitations of their study. The authors should underline both the advantages and disadvantages of their proposed approach when compared with other valuable studies from the current state of art. When discussing their obtained results, the authors should emphasize not only the novel aspects and strong points of their developed method, but also should point out objectively the existing limitations of their method, possible circumstances that will hinder their method’s effectiveness.

18. The "Experiment and analysis" section. The paper will benefit if the authors present more details regarding the results obtained during various tests, for all the different number of hidden layers, neurons and epochs tested and especially the training time for each test, until they have obtained the configuration that has provided the best results. The information can be summarized in a table and if it becomes too long, the authors can restrict it in the paper to ten main experimental runs, and a complete table with all the experimental runs must be inserted as an Annex of the article. How often does the network need to be retrained/updated and how did the authors tackle the need of retraining/updating the network? How is the data encountered stored for subsequent updates of the network?

19. Insights. The paper will benefit if the authors make a step further, beyond their approach and provide an insight when discussing their obtained results regarding what they consider to be, based on the obtained results, the most important benefits of the research conducted within the manuscript, taking also into account its practical applicability.

20. The cost-benefit analysis. It will benefit the paper if the authors elaborate a cost-benefit analysis regarding the implementation of their proposed solution in a real working environment, taking also into account the licensing cost of the software.

Other remarks.

·  Comments regarding the sections of the manuscript. According to the "Instructions for Authors" from the Sensors MDPI Journal's website (https://www.mdpi.com/journal/sensors/instructions), "Sensors now accepts free format submission: We do not have strict formatting requirements, but all manuscripts must contain the required sections: Author Information, Abstract, Keywords, Introduction, Materials & Methods, Results, Conclusions, Figures and Tables with Captions, Funding Information, Author Contributions, Conflict of Interest and other Ethics Statements."). The manuscript under review will benefit if it is restructured in accordance with the above-mentioned template that provides a more logical structure that is much more appropriate for a research article. The restructuring of the manuscript will also help the authors to better express the novelty of their work and the contribution that they have made to the current state of knowledge.

·   Run-on expressions. At Lines 143-144, the authors state: "The most representative algorithms are OCSVM [19], SVDD [5], etc." In a scientific paper one should avoid using run-on expressions, such as "and so forth", "and so on" or "etc.". Therefore, instead of "etc.", the sentences should mention all the elements that are relevant to the analysis and to the obtained results in order to obtain a consistent, relevant and reproductible study.

·   Line 6: "slicing.Nevertheless"Line 7: "intrusion.To solve", Line 15: "techniques.The experimental"Line 27: "control.Among". In order to assure the legibility for the readers, after using punctuation signs, a white space is necessary. This problem appears numerous times along the article.

·    I consider that the article under review will benefit if the authors carefully review and rewrite it more clearly, more accurate, as its current form contains a lot of weak and insufficiently detailed sentences, many unclear, inappropriate chosen expressions and assertions, lacking scientific soundness, for example: at Line 35"System components of the application plane include application applications…"., at Line 301"In this paper the model proposed in this paper…"

Author Response

Dear 

Thank you very much for your sincere suggestions for our manuscript.The detailed response is in the attachment.

Best regards,

Round 2

Reviewer 1 Report

The authors have addressed the concerns expressed.

Author Response

Dear Editor,

Thank you very much for your advice to us, your advice is very important to our thesis.

Yours Sincerely,

Reviewer 2 Report

After having assessed the suitability for publication of the revised version of the Manuscript ID: sensors-1905353, having the title "An anomaly detection algorithm based on ensemble learning for 5G environment", I have concluded the same aspects as the authors did, and I quote:

1. From the authors' point-by-point responses to my comments:

·       "At the same time, our laboratory does not have the physical conditions to carry out such experiments, and it is difficult for us to carry out relevant experimental verifications."

·       "In our experiment and analysis, we emphasized that our method is performed in a laboratory environment, not a real network environment, so our model does not necessarily have the ability to be applied in a real environment. And our method is computationally expensive due to the introduction of a large number of additional computations."

·       "Since our method is carried out in a laboratory environment, it is not necessarily applicable in a real environment."

2. From the revised version of the Manuscript ID: sensors-1905353:

·       "Since the experimental environment of the method in this paper is not the real environment, the method in this paper is not necessarily suitable for the real environment."

Therefore, I consider that the authors should address the issues that they themselves have identified, by carrying out relevant experimental tests according to a sound research methodology in order to obtain a method that can actually be used for the stated purpose, namely anomalies detection for the 5G environment.

Author Response

Dear Editor,

Thank you very much for your suggestion, your suggestion is very important to us. We have made changes to your suggestion. See the attachment for details of the revisions.

Yours Sincerely,

Round 3

Reviewer 2 Report

After having assessed the suitability for publication of the second revised version of the Manuscript ID: sensors-1905353, having the title "An anomaly detection algorithm based on ensemble learning for 5G environment", I can conclude that the authors have improved the manuscript.